# MetaQuant: Learning to Quantize by Learning to Penetrate Non-differentiable Quantization

**Shangyu Chen**
Nanyang Technological University, Singapore
schen025@e.ntu.edu.sg

**Wenya Wang**
Nanyang Technological University, Singapore
wangwy@ntu.edu.sg

**Sinno Jialin Pan**
Nanyang Technological University, Singapore
sinnopan@ntu.edu.sg

## Abstract

Tremendous amount of parameters make deep neural networks impractical to be deployed for edge-device-based real-world applications due to the limit of computational power and storage space. Existing studies have made progress on learning quantized deep models to reduce model size and energy consumption, i.e. converting full-precision weights ($r$'s) into discrete values ($q$'s) in a supervised training manner. However, the training process for quantization is non-differentiable, which leads to either infinite or zero gradients ($g_r$) w.r.t. $r$. To address this problem, most training-based quantization methods use the gradient w.r.t. $q$ ($g_q$) with clipping to approximate $g_r$ by Straight-Through-Estimator (STE) or manually design their computation. However, these methods only heuristically make training-based quantization applicable, without further analysis on how the approximated gradients can assist training of a quantized network. In this paper, we propose to learn $g_r$ by a neural network. Specifically, a meta network is trained using $g_q$ and $r$ as inputs, and outputs $g_r$ for subsequent weight updates. The meta network is updated together with the original quantized network. Our proposed method alleviates the problem of non-differentiability, and can be trained in an end-to-end manner. Extensive experiments are conducted with CIFAR10/100 and ImageNet on various deep networks to demonstrate the advantage of our proposed method in terms of a faster convergence rate and better performance. Codes are released at: https://github.com/csyhhu/MetaQuant

## 1 Introduction

Deep neural networks have shown promising results in various computer vision tasks. However, modern deep learning models usually contain many layers and enormous amount of parameters [9], which limits their applications on edge devices. To reduce parameters redundancy, continuous effects in architecture refinement have been made, such as using small kernel convolutions [14] and reusing features [6]. Consider a very deep model which is fully-trained. To use it for making predictions, most of the computations involve multiplications of a real-valued weight by a real-valued activation in a forward pass. These multiplications are expensive as they are all float-point to float-point multiplication operations. To alleviate this problem, a number of approaches have been proposed to quantize deep models. Courbariaux *et al.* [4] and Hubara *et al.* [7] proposed to binarize weights of the deep model to be in $\{\pm1\}$. To provide more flexibility for quantized values in each layer, Rastegari *et al.* [13] introduced a float value $\alpha_l$ known as the *scaling factor* for layer $l$ to turn binarized weights

into $\alpha_l \times \{\pm 1\}$. Li *et al.* [11] extended binary weights to ternary values, and Zhou *et al.* [17] further incorporated activation and gradient quantization.

Training-based quantization methods generate quantized neural networks under the training mechanism. Existing training-based quantization methods can be roughly categorized into "STE" and "Non-STE" methods. "STE" methods contain a non-differentiable discrete quantization function, connecting the full-precision weights and quantized weights. During backpropagation, STE is used to penetrate this non-differentiable function. (e.g.[7], [13], [17]). "Non-STE" methods are referred to as learning without STE by directly working on full-precision weights with a regularizer to obtain feasible quantization ([2]) or weights projection using proximal gradient methods ([10], [5]). The training process in Non-STE quantization suffers from heavy hyper-parameters tuning, such as weights partition portion in each step [15] and penalty setting in [10].

Specifically, STE quantization methods follow a rather simple and standard training protocol. Given a neural network $f$ with full-precision weights $\mathbf{W}$, a quantization function $Q(\cdot)$ (without loss of generalization, $Q(r)$ is set as a mapping from $r$ to 1 if $r \geq 0$, otherwise $-1$), and labeled data $(\mathbf{x}, y)$, the objective is to minimize the training loss: $\ell(f(Q(\mathbf{W}); \mathbf{x}), y)$. However, due to the non-differentiability of $Q$, the gradient of $\ell$ w.r.t $\mathbf{W}$ cannot be computed using the chain rule: $\frac{\partial \ell}{\partial \mathbf{W}} = \frac{\partial l}{\partial Q(\mathbf{W})} \frac{\partial Q(\mathbf{W})}{\partial \mathbf{W}}$, where $\frac{\partial Q(\mathbf{W})}{\partial \mathbf{W}}$ is infinite when $\mathbf{W} = 0$ and 0 elsewhere. To enable a stable quantization training, Hubara *et al.* [7] proposed Straight-Through-Estimator (STE) to redefine $\frac{\partial Q(r)}{\partial r}$:

$$\frac{\partial Q(r)}{\partial r} = \begin{cases} 1 & \text{if} & |r| \leq 1, \\ 0 & \text{otherwise.} \end{cases}.$$

STE is widely used in training-based quantization methods[1] as it provides an approximated gradient for penetration of $Q$ with an simple implementation. However, it inevitably brings the problem of *gradient mismatch*: the gradients of the weights are not generated using the value of weights, but rather its quantized value. Although STE provides an end-to-end training benifit under discrete constraints, few works have progressed to investigate how to obtain better gradients for quantization training. In the methods HWGQ [3] and Bi-real [12], $\frac{\partial Q(r)}{\partial r}$ is manually defined, but they focused on activation quantization.

To overcome the problem of gradient mismatch and explore better gradients in training-based methods, inspired by [1], we propose to learn $\frac{\partial Q(\mathbf{W})}{\partial \mathbf{W}}$ by a neural network ($\mathcal{M}$) in quantization training. This additional neural network is referred to as *meta quantizer* and trained together with the base quantized model. The whole process is denoted by **Meta Quant**ization (MetaQuant). Specially, in each backward propagation, $\mathcal{M}$ takes $\frac{\partial \ell}{\partial Q(\mathbf{W})}$ and $\mathbf{W}$ as inputs in a coordinate-wise manner, then its output is used to compute $\frac{\partial \ell}{\partial \mathbf{W}}$ for updating weights $\mathbf{W}$ using common optimization methods such as SGD or Adam [8]. In a forward pass, inference is performed using the quantized version of the updated weights, which produces the final outputs to be compared with the ground-truth labels for backward computation. During this process, gradient propagation from the quantized weights to the full-precision weights is handled by $\mathcal{M}$, which avoids the problem of non-differentiability and gradient mismatch. Besides, the gradients generated by the meta quantizer are loss-aware, contributing to better performance of the quantization training.

Compared with commonly-used STE and manually designed gradient propagation in quantization training, MetaQuant learns to generate proper gradients without any manually designed knowledge. The whole process is end-to-end. meta quantizer can be viewed as a plug-in to any base model, making it easy and general to be implemented in modern architectures. After quantization training is finished, meta quantizer can be removed and consumes no extra space for inference. We compare MetaQuant with STE under different quantization functions (dorefa [17], BWN [13]) and optimization techniques (SGD, Adam) with CIFAR10/100 and ImageNet on various base models to verify MetaQuant's generalizability. Extensive experiments show that MetaQuant achieves a faster convergence speed under SGD and better performance under SGD/Adam.

## 2 Related Work

Courbariaux *et al.* [4] proposed to train binarized networks through deterministic and stochastic rounding on parameters update after backpropagation. This idea was further extended in [7] and [13] by introducing binary activation. Nevertheless, these pioneer attempts face the problem of non-differentiable rounding operator during back-propagation, which is solved by directly penetration of rounding with unchanged gradient. To bypass non-differentiability, Leng *et al.* [10] modified the quantization training objective function using ADMM, which separates the processes on training real-valued parameters and quantizing the updated parameters. Zhou *et al.* [15] proposed to incrementally quantize a portion of parameters based on weight partition and update the un-quantized parameters by normal training. However, this kind of methods introduced more hyper-parameters tuning such as determining the procedure of partial quantization, thus complicating quantization. Bai *et al.* [2] added a regularizer in quantization training to transform full-precision weights to quantized values. Though this method simplifies quantization training procedure, but its optimization process involves the proximal method, which makes the training cost expensive.

## 3 Problem Statement

Given a training set of $n$ labeled instances $\{\mathbf{x}, y\}$'s, a pre-trained full-precision base model $f$ with $L$ layers is parameterized by $\mathbf{W} = [\mathbf{W}_1, ..., \mathbf{W}_L]$. We define a pre-processing function $\mathcal{A}(\cdot)$ and a quantization function $Q(\cdot)$. $\mathcal{A}(\cdot)$ converts $\mathbf{W}$ into $\tilde{\mathbf{W}}$, which is rescaled and centralized to make it easier for quantization. $Q(\cdot)$ discretizes $\tilde{\mathbf{W}}$ to $\hat{\mathbf{W}}$ using $k$-bits. Specially, 2 pre-processing functions and corresponding quantization methods (dorefa[2], BWN) are studied in this work:

$$\textbf{dorefa}: \ \tilde{\mathbf{W}} = \mathcal{A}(\mathbf{W}) = \frac{\tanh(\mathbf{W})}{2\max(|\tanh(\mathbf{W})|)} + \frac{1}{2}, \ \ \hat{\mathbf{W}} = Q(\tilde{\mathbf{W}}) = 2\frac{\text{round}\left[(2^k - 1)\tilde{\mathbf{W}}\right]}{2^k - 1} - 1. \ (1)$$

$$\textbf{BWN}: \ \tilde{\mathbf{W}} = \mathcal{A}(\mathbf{W}) = \mathbf{W}, \ \ \ \ \ \ \ \hat{\mathbf{W}} = Q(\tilde{\mathbf{W}}) = \frac{1}{n}||\tilde{\mathbf{W}}||_{l_1} \times \text{sign}(\tilde{\mathbf{W}}). \ (2)$$

Training-based quantization aims at training a quantized version of $\mathbf{W}$, i.e., $\hat{\mathbf{W}}$, such that the loss of the quantized $f$ is minimized: $\min_{\hat{\mathbf{W}}} \ell(f(\hat{\mathbf{W}}; \mathbf{x}), y)$.

## 4 Meta Quantization

### 4.1 Generation of Meta Gradient

Our proposed MetaQuant incorporates a shared meta quantizer $\mathcal{M}_\phi$ parameterized by $\phi$ across layers into quantization training. After $\mathbf{W}$ is quantized as $\hat{\mathbf{W}}$ (subscript $l$ is omitted for ease of notation), a loss $\ell$ is generated by comparing $f(\hat{\mathbf{W}}; \mathbf{x})$ with the ground-truth.

In back-propagation, the gradient of $\ell$ w.r.t $\hat{\mathbf{W}}$ is then computed by chain rules, which is denoted by $g_{\hat{\mathbf{W}}} = \frac{\partial \ell}{\partial \hat{\mathbf{W}}}$. The meta quantizer $\mathcal{M}_\phi$ receives $g_{\hat{\mathbf{W}}}$ and $\tilde{\mathbf{W}}$ as inputs, and outputs the gradient of $\ell$ w.r.t. $\tilde{\mathbf{W}}$, denoted by $g_{\tilde{\mathbf{W}}}$, as:

$$g_{\tilde{\mathbf{W}}} = \frac{\partial \ell}{\partial \tilde{\mathbf{W}}} = \mathcal{M}_\phi(g_{\hat{\mathbf{W}}}, \tilde{\mathbf{W}}). \ \ (3)$$

The gradient $g_{\tilde{\mathbf{W}}}$ is further used to compute the gradient of $\ell$ w.r.t. $\mathbf{W}$, denoted by $g_{\mathbf{W}}$, where $g_{\mathbf{W}}$ is computed via:

$$g_{\mathbf{W}} = \frac{\partial \ell}{\partial \tilde{\mathbf{W}}} \frac{\partial \tilde{\mathbf{W}}}{\partial \mathbf{W}} = g_{\tilde{\mathbf{W}}} \frac{\partial \tilde{\mathbf{W}}}{\partial \mathbf{W}} = \mathcal{M}_\phi(g_{\hat{\mathbf{W}}}, \tilde{\mathbf{W}}) \frac{\partial \tilde{\mathbf{W}}}{\partial \mathbf{W}}, \ \ (4)$$

where $\frac{\partial \tilde{\mathbf{W}}}{\partial \mathbf{W}}$ depends on the pre-processing function between $\mathbf{W}$ and $\tilde{\mathbf{W}}$: $\frac{\partial \tilde{\mathbf{W}}}{\partial \mathbf{W}} = \frac{1 - \tanh^2(\mathbf{W})}{\max(|\tanh(\mathbf{W})|)}$ for dorefa according to (1), and $\frac{\partial \tilde{\mathbf{W}}}{\partial \mathbf{W}} = \mathbf{1}$ for BWN according to (2). This process is referred to as *calibration*.

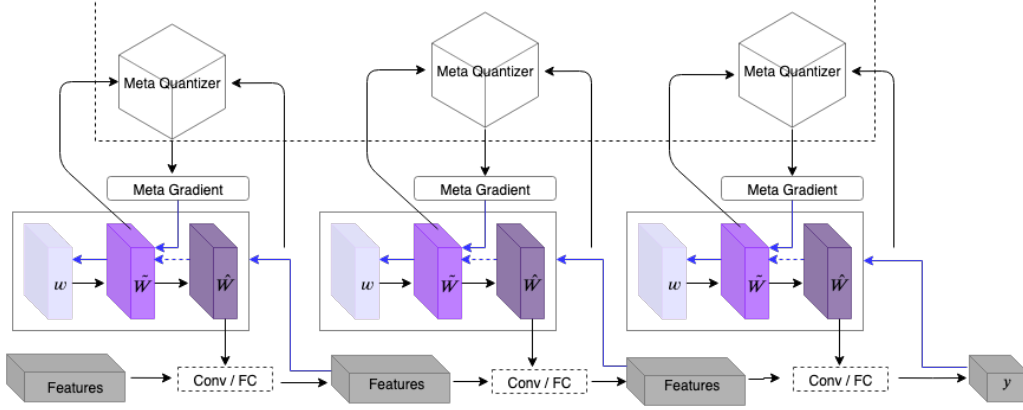

Figure 1: The overflow of MetaQuant. During backward propagation, gradients are represented as blue line. Dash blue line means this propagation is non-differentiable and requires special handling. A shared meta network $\mathcal{M}$ is constructed which takes $g_{\hat{\mathbf{W}}}$ and $\tilde{\mathbf{W}}$ as input, and outputs the gradient of $\tilde{\mathbf{W}}$ ($g_{\tilde{\mathbf{W}}}$). With $g_{\tilde{\mathbf{W}}}$, the gradient of the weights $\mathbf{W}$ can be computed using (4). Finally, $\mathbf{W}$ is updated with (5), with the assistance of different optimization methods reflected in $\pi(\cdot)$.

Before using $g_{\mathbf{W}}$ to update $\mathbf{W}$, $g_{\mathbf{W}}$ is firstly processed according to different optimization methods to produce the final update value for each weight. This process is named *gradient refinement*, which is denoted by $\pi(\cdot)$ in the sequent. Specifically, for SGD, $\pi(g_{\mathbf{W}}) = g_{\mathbf{W}}$. For other optimization methods such as Adam, $\pi(\cdot)$ can be implemented as $\pi(g_{\mathbf{W}}) = g_{\mathbf{W}} + residual$, where "residual" is computed according to different gradient refinement methods. Finally, the full-precision weights $\mathbf{W}$ is updated as:

$$\mathbf{W}^{t+1} = \mathbf{W}^t - \alpha \pi(g_{\mathbf{W}}^t), \tag{5}$$

where $t$ denotes the $t$-th training iteration and $\alpha$ is the learning rate. Fig.1 illustrates the overall procedure of MetaQuant.

Compared with [1], which directly learns $g_{\mathbf{W}}$, MetaQuant construct a neural network to learn $g_{\tilde{\mathbf{W}}}$, which cannot be directly computed in quantization training due to the property of non-differentiability of the quantization functions. Our work resolves the issue of non-differentiability and is general to different optimization methods. Insight of how and why MetaQuant works is studied at Appendix.7.2.

## 4.2 Training of Meta Quantizer

Similar to [1], our proposed meta quantizer is a **coordinate-wise** neural network, which means that each weight parameter is processed independently. For a *single* weight index $i$ in $g_{\hat{\mathbf{W}}_i}$, $\tilde{\mathbf{W}}_i$ receives its corresponding gradient $g_{\tilde{\mathbf{W}}_i}$ via $g_{\tilde{\mathbf{W}}_i} = \mathcal{M}_\phi(g_{\hat{\mathbf{W}}_i}, \tilde{\mathbf{W}}_i)$. For efficient processing, during inference, the inputs in (3) are arranged as batches with size 1. Specially, suppose $\mathbf{W}$ comes from a convolution layer with shape $\mathbb{R}^{o \times i \times k \times k}$, where $o$, $i$ and $k$ denote the number of output channels, input channels and kernel size, respectively. Then $\tilde{\mathbf{W}}$, $\hat{\mathbf{W}}$ and the corresponding gradient share the same shape, which is a reshaping of inputs in (3) to $\mathbb{R}^{(o \times i \times k^2) \times 1}$.

Recall from (5) and (4), the output of $\mathcal{M}_\phi$ is incorporated into the value of updated $\mathbf{W}^t$, which is then quantized in next iteration's inference.

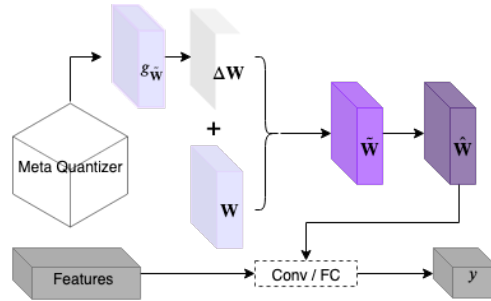

Figure 2: Incorporation of meta quantizer into quantization training. $\Delta \mathbf{W}$ is composed of calibration, gradient refinement and multiplication of learning rate $\alpha$. Output of meta quantizer is involved in $\mathbf{W}$'s update and contributes to final loss, constructing a differential path from loss to $\phi$-parameterized meta quantizer.

Therefore, $\mathcal{M}_\phi$ is associated to the final quantization training loss, which receives gradient update on $\phi$ backpropagated from the final loss. By introducing the meta quantizer to produce $g_{\tilde{\mathbf{W}}}$, MetaQuant not only addresses the non-differentiability issue for parameters in the base model, but also provides an end-to-end training benefit throughout the whole network. Moreover, the meta quantizer is loss-aware, hence it is trained to generate more accurate update for $\mathbf{W}$ for reducing the final loss, which explores how gradient can be modified to assist quantization training. Figure.2 illustrates the detailed process when incorporating the meta quantizer into the quantization training of the base model, which forms a differentiable path from the final loss to $\phi$. In the meantime of quantization training in $\mathbf{W}$, $\phi$ is also learned in each training iteration $t$:

$$\text{Forward:} \quad \tilde{\mathbf{W}}^t = \mathcal{A}(\mathbf{W}^t) = \mathcal{A}\left[\mathbf{W}^{t-1} - \alpha \times \pi(\mathcal{M}_\phi(g_{\hat{\mathbf{W}}}^{t-1}, \tilde{\mathbf{W}}^{t-1})\frac{\partial \tilde{\mathbf{W}}^{t-1}}{\partial \mathbf{W}^{t-1}})\right], \quad (6)$$

$$\text{Loss} = \ell\left(f\left[Q(\tilde{\mathbf{W}}^t); \mathbf{x}\right], y\right), \quad (7)$$

$$\text{Backward:} \quad \frac{\partial \ell}{\partial \phi^t} = \frac{\partial \ell}{\partial \tilde{\mathbf{W}}^t}\frac{\partial \tilde{\mathbf{W}}^t}{\partial \phi^t} = \mathcal{M}_\phi(g_{\hat{\mathbf{W}}^t}, \tilde{\mathbf{W}}^t)\frac{\partial \tilde{\mathbf{W}}^t}{\partial \phi^t}. \quad (8)$$

In Forward, we use a combination of $\mathbf{W}^{t-1}$ and meta gradient to represent $\mathbf{W}^t$, in order to incorporate $\mathcal{M}_\phi$. Specially in (6), meta gradient is derived from $\mathcal{M}$'s output, which is firstly multiplied to achieve gradient of $\mathbf{W}$, then is refined by optimization $\pi$. Finally, it is adjusted by learning rate to become meta gradient. (8) calculates gradient of $\phi$, here $\frac{\partial \tilde{\mathbf{W}}}{\partial \phi}$ is differentiable because $\mathcal{A}$ is differentiable. Furthermore, a differentiable meta neural network is chosen. $\mathbf{W}_t$ will be actually updated after Backward, which can be regarded as late weights update.

## 4.3 Design of Meta Quantizer

The meta quantizer $\mathcal{M}_\phi$ is a parameterized and differentiable neural network to generate the meta gradient. It can be viewed as a generalization of STE. For example, $\mathcal{M}_\phi$ reduces to STE if it clips $g_{\tilde{\mathbf{W}}}$ according to the absolute magnitude of $\tilde{\mathbf{W}}$: $g_{\tilde{\mathbf{W}}} = \mathcal{M}_\phi(g_{\hat{\mathbf{W}}}, \tilde{\mathbf{W}}) = g_{\hat{\mathbf{W}}} \cdot \mathbf{1}_{|\tilde{\mathbf{W}}| \leq 1}$.

We design 3 different architectures of the meta quantizer. The first architecture simply uses a neural network composing of 2 or multiple layers of fully-connected layer. It only requires $g_{\hat{\mathbf{W}}}$ as input:

$$\texttt{FCGrad}: \qquad \mathcal{M}_\phi(g_{\hat{\mathbf{W}}}) = \text{FCs}(\phi, \sigma, g_{\hat{\mathbf{W}}}), \qquad (9)$$

where $\sigma$ represents the nonlinear activation. Since previous successful experimental results brought by STE show that a good $g_{\tilde{\mathbf{W}}}$ should be generated by considering the value of $\tilde{\mathbf{W}}$. Based on this observation, we construct another 2 architectures of meta quantizer with $\tilde{\mathbf{W}}$ fed as input and multiply the output of these neural networks with $g_{\hat{\mathbf{W}}}$ to incorporate gradient information from its subsequent step. Specifically, one is based on fully-connected (FC) layers:

$$\texttt{MultiFC}: \qquad \mathcal{M}_\phi(g_{\hat{\mathbf{W}}}, \tilde{\mathbf{W}}) = g_{\hat{\mathbf{W}}} \cdot \text{FCs}(\phi, \sigma, \tilde{\mathbf{W}}). \qquad (10)$$

Another network incorporates LSTM and FC to construct $\mathcal{M}$, which is inspired by [1] that uses memory-based neural network as the meta learner:

$$\texttt{LSTMFC}: \qquad \mathcal{M}_\phi(g_{\hat{\mathbf{W}}}, \tilde{\mathbf{W}}) = g_{\hat{\mathbf{W}}} \cdot \text{FCs}(\phi_{FCs}, \sigma, (\text{LSTM}(\phi_{LSTM}, \tilde{\mathbf{W}}))). \qquad (11)$$

When using LSTM as the meta quantizer, each coordinate of the weights keeps a track of the hidden states generated by LSTM, which contains the memory of historical information of $g_{\hat{\mathbf{W}}}$ and $\tilde{\mathbf{W}}$. Meta quantizer's memory consumption and detailed hyper-parameter is studied at Appendix.7.1, 7.3.

## 4.4 Algorithm and Implementation Details

The detailed process of MetaQuant is illustrated in Algorithm 1. A shared meta quantizer $\mathcal{M}_\phi$ is firstly constructed and randomly initialized. During each training iteration, line 2-6 describes the forward process: for each layer, $g_{\hat{\mathbf{W}}}$ and $\tilde{\mathbf{W}}$ from the previous iteration are fed into $\mathcal{M}_\phi$ to generate the meta gradient $g_{\tilde{\mathbf{W}}}$ to perform inference, as indicated from line 3-5. Since $g_{\hat{\mathbf{W}}}$ is not calculated in the first iteration, normal quantization training is conducted at the first iteration: $\hat{\mathbf{W}} = Q(\tilde{\mathbf{W}}) = Q\left[\mathcal{A}(\mathbf{W})\right]$ to replace line 4. Line 7-9 shows the backward process: $\hat{\mathbf{W}}$'s gradient can be attained through

error backpropagation, which is shown in line 7. During the backward process, $g_{\hat{\mathbf{W}}}$ and $\tilde{\mathbf{W}}$ of the current iteration are obtained and their outputs from $\mathcal{M}_\phi$ are saved for computation in the next iteration, denoted by $g_{\tilde{\mathbf{W}}^{t+1}}$ as described in line 7-8. By incorporating $\mathcal{M}_\phi$ into the inference graph, its gradient is obtained in line 9. Finally, $g_{\tilde{\mathbf{W}}}$ is used to calculate $g_{\mathbf{W}}$, which is then processed by different optimization methods using $\pi(\cdot)$, leading to the update of $\mathbf{W}$ shown in line 10-12. In the first iteration, due to the lack of $g_{\tilde{\mathbf{W}}}$, weights update of $\mathbf{W}$ is not conducted. Note that $\phi$ from the meta quantizer is updated in line 13.

---

**Algorithm 1** MetaQuant

---

**Require:** Training dataset $\{\mathbf{x}, y\}^n$, well-trained full-precision base model $\mathbf{W}$.
**Ensure:** Quantized base model $\hat{\mathbf{W}}$.
1: Construct shared meta quantizer $\mathcal{M}_\phi$, training iteration $t = 0$.
2: **while** not optimal **do**
3:     **for** Layer $l$ from 1 to $L$ **do**
4:         $\hat{\mathbf{W}}_l^t = Q(\tilde{\mathbf{W}}_l^t) = Q\left\{\mathcal{A}\left[\mathbf{W}_l^{t-1} - \alpha \times \pi(\mathcal{M}_\phi(g_{\hat{\mathbf{W}}_l}^{t-1}, \tilde{\mathbf{W}}_l^{t-1}) \cdot \frac{\partial \tilde{\mathbf{W}}_l^{t-1}}{\partial \mathbf{w}_l^{t-1}})\right]\right\}$
5:     **end for**
6:     Calculate loss: $\ell = \text{Loss}\left\{f\left[Q(\tilde{\mathbf{W}}^t); \mathbf{x}\right], y\right\}$
7:     Generate $g_{\hat{\mathbf{W}}^t}$ using chain rules.
8:     Calculate meta gradient $g_{\tilde{\mathbf{W}}^t}$ using $\mathcal{M}_\phi$.
9:     Calculate $\frac{\partial \ell}{\partial \phi^t}$ by (8)
10:    **for** Layer $l$ from 1 to $L$ **do**
11:       $\mathbf{W}_l^t = \mathbf{W}_l^{t-1} - \alpha \times \pi(\mathcal{M}_\phi(g_{\hat{\mathbf{W}}_l}^{t-1}, \tilde{\mathbf{W}}_l^{t-1}) \cdot \frac{\partial \tilde{\mathbf{W}}_l^{t-1}}{\partial \mathbf{w}_l^{t-1}})$
12:    **end for**
13:    $\phi^{t+1} = \phi^t - \gamma \times \frac{\partial \ell}{\partial \phi^t}$ ($\gamma$ is the learning rate of the meta quantizer)
14:    $t = t + 1$
15: **end while**

---

# 5 Experiment

## 5.1 Experiment Setup

MetaQuant focuses on the penetration of non-differentiable quantization function during training-based methods. We conduct comparison experiments with STE under the following 2 forward quantization methods: 1) dorefa [17] , 2) BWN [13] and 2 optimization methods: 1) SGD 2) Adam [8]. When quantization training is conducted with dorefa or BWN as forward quantization function and STE as backward method, it becomes a weight-quantization version of [17] or the proposed method in [13], respectively. Three benchmark datasets are used including ImageNet ILSVRC-2012 and CIFAR10/100. Regarding deep architectures, we experiment with ResNet20/32/44 on CIFAR10. Since CIFAR10/100 share the same input dimension, we modify the output dimension of the last fully-connected layer from 10 to 100 in ResNet56/110 for CIFAR100. For ImageNet, ResNet18 is utilized for comparison. For all the experiments conducted and compared, **all** layers in the networks are quantized using **1 bit**: each layer contains only 2 values. For experiments on CIFAR10/100, we set the initial learning rate as $\alpha = 1e^{-3}$ for base models and the initial learning rate as $\gamma = 1e^{-3}$ for the meta quantizer. For fair comparison, we set total training epochs as 100 for all experiments, $\alpha$ and $\gamma$ will be divided by 10 after every 30 epochs. For ImageNet, the initial learning rate is set as $\alpha = 1e^{-4}$ for the base model using dorefa and BWN. Initial $\gamma$ is set as $1e^{-3}$. $\alpha$ decreases to $\{1e^{-5}, 1e^{-6}\}$ when training comes to 10 / 20 epochs. $\gamma$ reduces to $\{1e^{-4}, 1e^{-5}\}$ in accordance to the change of the learning rate in base models with total epoch as 30. Batch size is 128 for CIFAR/ImageNet. All experiments are conducted for 5 times, the statistics of last 10/5 epochs' test accuracy are reported as the performance of both proposed and baseline methods in CIFAR/ImageNet datasets. We also demonstrate the empirical convergence speed among different methods through training loss curves.

Detailed hyper-parameters in different realizations of MetaQuant in CIFAR experiments are the following: In `MultiFC`, a 2-layer fully-connected layer is used with hidden size as 100, no non-linear activation is used. In `LSTMFC`, a 1-layer LSTM and a fully-connected layer are utilized, with the hidden dimension set as 100. In `FCGrad`, a 2-layer fully-connected meta model is used with hidden

size as 100 without non-linear activation. In ImageNet experiments, we use `MultiFC/FCGrad` with 2/1-layer fully-connected layer, whose hidden dimension is 100.

## 5.2 Experimental Results and Analysis

| Network | Forward | Backward | Optimization | Test Acc (%) | FP Acc (%) |
|---|---|---|---|---|---|
| ResNet20 | dorefa | STE | SGD | 80.745(2.113) | 91.5 |
| | | MultiFC | | **88.942(0.466)** | |
| | | LSTMFC | | 88.305(0.810) | |
| | | FCGrad | | 88.840(0.291) | |
| | | STE | Adam | 89.782(0.172) | |
| | | MultiFC | | 89.941(0.068) | |
| | | LSTMFC | | **89.979(0.103)** | |
| | | FCGrad | | 89.962(0.068) | |
| | BWN | STE | SGD | 75.913(3.495) | |
| | | LSTMFC | | **89.289(0.212)** | |
| | | FCGrad | | 88.949(0.231) | |
| | | STE | Adam | 89.896(0.182) | |
| | | LSTMFC | | 90.036(0.109) | |
| | | FCGrad | | **90.042(0.098)** | |
| ResNet32 | dorefa | STE | SGD | 82.911(1.680) | 92.13 |
| | | MultiFC | | 89.637(0.380) | |
| | | LSTMFC | | **90.397(0.149)** | |
| | | FCGrad | | 89.934(0.246) | |
| | | STE | Adam | 90.172(0.077) | |
| | | MultiFC | | 90.966(0.064) | |
| | | LSTMFC | | 90.948(0.074) | |
| | | FCGrad | | **90.976(0.068)** | |
| | BWN | STE | SGD | 79.768(2.062) | |
| | | LSTMFC | | **90.568(0.169)** | |
| | | FCGrad | | 90.241(0.316) | |
| | | STE | Adam | 91.015(0.087) | |
| | | LSTMFC | | 91.002(0.077) | |
| | | FCGrad | | **91.034(0.067)** | |
| ResNet44 | dorefa | STE | SGD | 86.686(1.020) | 93.56 |
| | | MultiFC | | 90.546(0.218) | |
| | | LSTMFC | | 91.494(0.163) | |
| | | FCGrad | | **91.539(0.097)** | |
| | | STE | Adam | 91.079(0.064) | |
| | | MultiFC | | 91.772(0.073) | |
| | | LSTMFC | | 91.870(0.022) | |
| | | FCGrad | | **91.989(0.067)** | |
| | BWN | STE | SGD | 82.647(0.334) | |
| | | LSTMFC | | 91.498(0.057) | |
| | | FCGrad | | **91.614(0.081)** | |
| | | STE | Adam | 91.121(0.023) | |
| | | LSTMFC | | 91.498(0.271) | |
| | | FCGrad | | **92.107(0.059)** | |

Table 1: Experimental result of MetaQuant and STE using dorefa, BWN on CIFAR10

Table.1 shows the overall experimental results on CIFAR10 for MetaQuant and STE using different forward quantization methods and optimizations. Variants of MetaQuant shows significant improvement over STE baseline, especially SGD is used.

CIFAR100 is a more difficult task than CIFAR10, which contains much more fine-grained classes with a total number of 100 classes. Table.2 shows the overall experimental results on CIFAR100 for MetaQuant and STE using different forward quantization methods and optimizations. Similar to CIFAR10, MetaQuant out-performs by a large margin than STE in all cases, showing that MetaQuant has significant improvement in more challenging tasks than traditional methods.

## 5.3 Empirical Convergence Analysis

In this experiment, we compare the performances of variants of MetaQuant and STE during the training process to demonstrate their convergence speeds. ResNet20 using dorefa is utilized as an example. As Fig.3 shows, under the same task and forward quantization method, MetaQuant shows tremendous convergence advantage over STE using SGD, including much faster descending speed of loss and obviously lower loss values. In Adam, although all the methods show similar decreasing

| Network | Forward | Backward | Optimization | Test Acc (%) | FP Acc (%) |
|---|---|---|---|---|---|
| ResNet56 | dorefa | STE | SGD | 42.265(8.143) | |
| | | MultiFC | | **65.791(0.415)** | |
| | | LSTMFC | | 63.645(2.183) | |
| | | FCGrad | | 64.351(0.935) | |
| | | STE | Adam | 66.419(0.533) | |
| | | MultiFC | | **66.588(0.375)** | |
| | | LSTMFC | | 66.483(0.793) | |
| | | FCGrad | | 66.564(0.351) | 71.22 |
| | BWN | STE | SGD | 34.479(11.737) | |
| | | LSTMFC | | 63.346(2.253) | |
| | | FCGrad | | **64.402(1.434)** | |
| | | STE | Adam | 64.297(1.309) | |
| | | LSTMFC | | 66.584(0.349) | |
| | | FCGrad | | **67.018(0.329)** | |
| ResNet110 | dorefa | STE | SGD | 43.419(18.902) | |
| | | MultiFC | | **68.269(0.136)** | |
| | | LSTMFC | | 64.753(2.850) | |
| | | FCGrad | | 66.145(2.490) | |
| | | STE | Adam | 66.836(1.198) | |
| | | MultiFC | | 68.418(0.235) | |
| | | LSTMFC | | 67.138(1.286) | |
| | | FCGrad | | **68.741(0.363)** | 72.54 |
| | BWN | STE | SGD | 35.227(19.408) | |
| | | LSTMFC | | **66.242(2.979)** | |
| | | FCGrad | | 64.791(4.096) | |
| | | STE | Adam | 66.265(1.429) | |
| | | LSTMFC | | 67.767(1.391) | |
| | | FCGrad | | **69.114(0.181)** | |

Table 2: Experimental result of MetaQuant and STE using dorefa, BWN on CIFAR100

| Network | Forward | Backward | Optimization | FP Top1/Top5(%) | Quant Top1/Top5 (%) |
|---|---|---|---|---|---|
| ResNet18 | dorefa | STE | Adam | 69.76/89.08 | 58.349(2.072)/81.477(1.567) |
| | | MultiFC | | | 59.472(0.025)/82.410(0.010) |
| | | FCGrad | | | **59.835(0.359)/82.671(0.232)** |
| | BWN | STE | | | 59.503(0.835)/82.549(0.506) |
| | | FCGrad | | | **60.328(0.391)/83.025(0.234)** |

Table 3: Experimental result of MetaQuant and STE using dorefa, BWN on ImageNet.

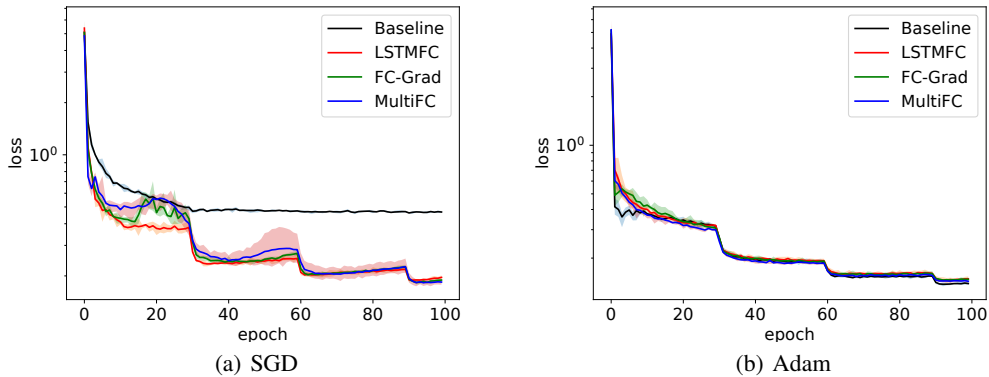

(a) SGD          (b) Adam

Figure 3: Convergence speed of MetaQuant V.S STE using SGD/Adam in ResNet20, CIFAR10, dorefa.

speed, MetaQuant methods finally reach to lower loss values, which is also reflected in the test accuracy reported in Table.1. Overall, MetaQuant shows better convergence than STE using different forward quantizations and optimizations. The improvement is more obvious when SGD is chosen.

We conjecture that the performance difference between SGD and Adam is due to the following reason: SGD simply updates full-precision weights using the calibrated gradient from $g_{\tilde{\mathbf{W}}}$, which directly reflects the output of meta quantizercompared to STE. Adam aggregates the historical information of $g_{\mathbf{W}}$ and normalizes the current gradient, which to a certain degree shrinks the difference of meta quantizer and STE. More comparisons in training accuracy, test accuracy on more tasks are listed in Appendix.7.4.

### 5.4 Performance Comparison with Non-STE Training-based Quantization

| Network | Method | Acc Drop (%) | Network | Method | Acc Drop (%) |
|---|---|---|---|---|---|
| ResNet20 | ProxQuant | 1.29 | ResNet32 | ProxQuant | 1.28 |
|  | MetaQuant | **0.7** |  | MetaQuant | **0.39** |
| ResNet44 | ProxQuant | 0.99 | LABNet | LAB | 1.4 |
|  | MetaQuant | **0.08** |  | MetaQuant | **-0.2** |
| ResNet18 | ELQ | **3.55/2.65** | ResNet18-2bits | TTQ [18] | **3.00/2.00** |
|  | MetaQuant | 6.32/4.31 |  | MetaQuant | 5.17/3.59 |

Table 4: Experimental result of MetaQuant V.S ProxQuant, LAB, ELQ, TTQ.

MetaQuant aims at improving training-based quantization by learning better gradients for penetration of non-differentiable quantization functions. Some advanced quantization methods avoid discrete quantization. In this section, we compare MetaQuant with Non-STE training-based quantization: ProxQuant ([2]), LAB ([5]) to demonstrate that traditional STE training-based quantization is able to achieve better performance by using MetaQuant.

Due to the difference of the initial full-precision model used, we only report the performance drop in terms of test accuracy after quantization (the smaller the better). We compare MetaQuant with ProxQuant using ResNet20/32/44, LAB using its proposed architecture[3] on CIFAR10 with all layers quantized to binary values. As shown in Table.4, MetaQuant shows better performance than both baselines.

ELQ ([16]) and TTQ ([18]) are compared in 3rd row in Table.4 using ImageNet datasets. Although over-performance, ELQ is a combination of a series of previous quantization methods and tricks on incremental quantization. MetaQuant focuses more on how to improve STE-based training quantization, without any extra loss and training tricks. TTQ is a non-symmetric ternarization with $\{0, \alpha, -\beta\}$ as ternary points. MetaQuant follows dorefa using a symmetric quantization which leads to efficient inference.

### 5.5 MetaQuant Training Analysis

Training of MetaQuant involves computation in training of meta quantizer. To analyze the additional training time, training time per iteration as for MetaQuant using `MultiFC` and DoReFa with STE using ResNet20 in CIFAR10 (Intel Xeon CPU E5-1650 with GeForce GTX 750 Ti). MetaQuant costs **51.15** seconds to finish one iteration of training while baseline method uses **38.17s**. However, In real deployment meta quantizer is removed, MetaQuant is able to provide better test performance without any extra inference time.

## 6  Conclusion

In this paper, we propose a novel method (MetaQuant) to learn the gradient for penetration of the non-differentiable quantization function in training-based quantization by a meta quantizer. This meta network is general enough to be incorporated into various base models and can be updated using the loss of the base models. We propose 3 types of meta quantizer and show that the meta gradients generated through these modules are able to provide better convergence speed and final quantization performance, under different forward quantization functions and optimization methods.

## Acknowledgement

This work is supported by NTU Singapore Nanyang Assistant Professorship (NAP) grant M4081532.020, and Singapore MOE AcRF Tier-2 grant MOE2016-T2-2-06.

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

## Footnotes

[1]In the following description, training-based quantization refers to STE training-based quantization

[2] In this work, we only consider the forward quantization function for weights quantization used in [17], and denote it as "dorefa"

[3](2x128C3)-MP2-(2x256C3)-MP2-(2x512C3)-MP2-(2x1024FC)-10FC
