[Supplementary Material · MetaQuant-12-18.pdf]

# 7 Appendix

## 7.1 Memory Usage Analysis and Slim Version of MetaQuant

During training of MetaQuant, the meta quantizer is updated, which causes extra memory consumption that is related to the number of parameters in the base mode. Empirically, in ResNet-based models, extra memory occupies 10% of the total consumption. To deal with those base models with a large number of parameters or when the memory is limited, we propose a slim version of MetaQuant to reduce the extra memory brought by meta quantizer.

Specially, in each training iteration, only parameters from one layer, which is randomly selected, is fed into meta quantizer for training, and the rest is fed for mere inference and for generating the meta gradient. This slim version is called MetaQuant-Slim, and we compare it with non-slim version using ResNet20, CIFAR10, SGD, dorefa as an example to show how slimming affect the performance of MetaQuant.

| Method | meta quantizer | Quant Acc (%) |
|---|---|---|
| Slim | MultiFC | 88.7 |
| Non-Slim | | 88.05 |
| Slim | LSTMFC | 88.23 |
| Non-Slim | | 88.11 |
| Slim | FCGrad | 88.28 |
| Non-Slim | | 89.06 |

Table 5: Comparison of slim and origin verison of MetaQuant using 3 types of meta quantizer in ResNet20, CIFAR10, optimizer as SGD, forward quantization as dorefa.

As shown in Table.5, the slim version of MetaQuant doesn't effect much to the final performance, but even provide slight improvement in some cases.

## 7.2 Looking Deeper into the meta quantizer

This section investigates how MetaQuant actually accelerates training convergence and improves final performance. In FCGrad, if no non-linear activation is used, the 2-layer fully-connected meta quantizer is equivalent to an adaptive factor multiplied in $g_{\hat{\mathbf{W}}}$ as shown in (9). We studied the changing of this amplification factor with different number of hidden sizes in meta quantizer during training using ResNet20, CIFAR10, dorefa, SGD as an example. As shown in Fig.4, FCGrad magnifies $g_{\hat{\mathbf{W}}}$'s magnitude adaptively (shown in dotted lines) during training, in accordance with the descending of the training loss. Larger hidden dimensions bring higher increase of the amplification, leading to faster convergence. Finally, all 3 meta quantizer climb to similar performances. In the analysis of FCGrad, it shows that MetaQuant adaptively tunes $g_{\tilde{\mathbf{W}}}$ by a multiplication of $g_{\hat{\mathbf{W}}}$, which is able to produce faster convergence and better performance in training-based quantization.

## 7.3 Analysis of Hyperparameters and Architecture in Meta Quantizer

(10), (9), (11) describes the general architecture of meta quantizer. However, how to choose the proper network and hidden dimension is important. The sensitivity of these hyper-parameters affects the utility of MetaQuant. In this section, we take ResNet20 in CIFAR10 using dorefa, Adam as an example to demonstrate how hidden dimension, activation and number of layers affect performance in MultiFC. As Table.6 shows, after hidden dimension climbs up to 30, the variation of the hidden size doesn't affect the performance of MetaQuant. Besides, the activation shows similar effects to the final performance. These results show that MetaQuant is insensitive to the hyper-parameters once it has already been equipped with enough capacity for meta training.

Figure 4: Loss and gradient amplification during training.

| Hidden | Activation | Acc |
|--------|-----------|-------|
| 1 | | 89.91 |
| 30 | None | 90.48 |
| 100 | | 90.48 |
| 1500 | | 90.46 |
| 100 | ReLU | 90.49 |
| | tanh | 90.46 |

Table 6: Experimental result of `MultiFC` under different hidden dimension, activation

## 7.4 Completed Convergence Analysis

### 7.4.1 ResNet20-CIFAR10

Convergence is shown in Fig.5 and Fig.6.

(a) SGD　　　　　　　　　　　　　　　　(b) Adam

Figure 5: Convergence speed of MetaQuant V.S STE using SGD/Adam in ResNet20, CIFAR10, dorefa.

(a) SGD　　　　　　　　　　　　　　　　(b) Adam

Figure 6: Convergence speed of MetaQuant V.S STE using SGD/Adam in ResNet20, CIFAR10, BWN.

### 7.4.2 ResNet32-CIFAR10

Convergence is shown in Fig.7 and Fig.8.

### 7.4.3 ResNet44-CIFAR10

Convergence is shown in Fig.9 and Fig.10.

### 7.4.4 ResNet56-CIFAR100

Convergence is shown in Fig.11 and Fig.12.

### 7.4.5 ResNet110-CIFAR100

Convergence is shown in Fig.13 and Fig.14.

Figure 7: Convergence speed of MetaQuant V.S STE using SGD/Adam in ResNet32, CIFAR10, dorefa.

Figure 8: Convergence speed of MetaQuant V.S STE using SGD/Adam in ResNet32, CIFAR10, BWN.

Figure 9: Convergence speed of MetaQuant V.S STE using SGD/Adam in ResNet44, CIFAR10, dorefa.

Figure 10: Convergence speed of MetaQuant V.S STE using SGD/Adam in ResNet44, CIFAR10, BWN.

Figure 11: Convergence speed of MetaQuant V.S STE using SGD/Adam in ResNet56, CIFAR100, dorefa.

Figure 12: Convergence speed of MetaQuant V.S STE using SGD/Adam in ResNet56, CIFAR100, BWN.

(a) SGD

(b) Adam

Figure 13: Convergence speed of MetaQuant V.S STE using SGD/Adam in ResNet110, CIFAR100, dorefa.

(a) SGD

(b) Adam

Figure 14: Convergence speed of MetaQuant V.S STE using SGD/Adam in ResNet110, CIFAR100, BWN.

### 7.4.6 ResNet18-ImageNet

Convergence is shown in Fig.15

Figure 15: Convergence speed of MetaQuant V.S STE using Adam in ResNet18, ImageNet, dorefa.