[Reviews · NeurIPS 2019]

Reviewer 1



Most existing neural network quantization methods use STE when performing extremely low-bit quantization tasks such as binary and ternary ones. They assume the full-precision reference and the quantized model have the same loss gradient for easy implementation. This paper proposes MetaQuant, a really novel method to calculate more accurate gradients from the training perspective. The proposed MetaQuant bridges the Gq w.r.t. quantized weights and r of full-precision weights as the inputs to a meta network, which outputs Gr and is trained jointly with the classification network (needs to be quantized) in an end-to-end manner. Three designs of meta quantizer are provided, and validated on DoReFa-Net and BWN using different image classification datasets and settings. Generally, the paper is very well-written, including the motivation, the theoretical analysis, the proposed design, the practical implementation, and the experiment settings and results. Here, I have some questions: (1) In the current design, the meta quantizer is shared across all layers and each weight parameter is processed independently. As weight parameters in kernels/filters are correlated, have the author tried other design harnessing weight relations? By encoding weight correlations via better weights sharing designs for the meta quantizer, improved accuracy may be obtained. (2) The meta quantizer will introduce extra memory cost as partially described in the supp material, how about its impact to training time cost (not the number of iterations)? I suggest the author to put this part of experiments to the main paper. (3) I would like to see a more comprehensive comparison on ImageNet, e.g., including more state-of-the-art results on binary/ternary networks. ------------------------------------------------------------------------------------------------------ My questions are well addressed by the author responses. I think this paper is a decent submission, thus I retain the score of 7.

Reviewer 2



The main issues of the paper are in the training of the meta quantizer (most of which is discussed in section 4.2). - In eq. (8) the term \partial \tilde{W} / \partial \phi is needed in the backward phase. However, from the solution setup description provided by the authors, \tilde{W} occurs before \phi in the computation graph. I fail to see how \partial \tilde{W} / \partial \phi can thus be computed, or even what it represents. A clarification from the authors would be appreciated. Note, it could be that auto-differentiation does not crash when this gradient is called (and this could explain why the method runs) - however, that is not enough evidence, a deeper explanation on what the term means and how it is computed is required. - Still in eq. (8) there seems to be a chicken-egg problem. The term \partial L / \partial \tilde{W} is replaced by M(...), the output of the meta-quantizer. This means that the dependence on the loss function L is suppressed and the updates based on the gradients computed in eq. (8) do not in fact operate to minimize L. The authors should clarify how the meta-quantizer is linked to the loss function. - This brings me to my next point: should the loss function of the base network be used for training the meta-quantizer? This seems not to be very well thought about. The two networks have different tasks, and the meta-quantizer is a regressor. A convincing discussion is needed to address this issue which I think is the main weakness of the paper as it stands. - Finally, I would like to point out to the authors that there are many writing imprecisions that severely harm the quality of the paper. For instance, some symbols are utilized without being defined/introduced (e.g., the boldface 1). The notation is inconsistent: for instance L is used to denote both the number of layers and the loss function. There are some typographic mistakes, e.g., 'outperforms' does not require a hyphen, and the same applies to 'fully connected' etc.. Post Response Comments: I thank the authors for their response. The feedback from the authors has helped me better articulate my issue with the proposed method, which I believe is very serious. Indeed, in the rebuttal, the authors show at line 22 how the computation occurs: "phi -> delta W -> W tilde -> W hat -> L". Clearly from Figure 1 in the rebuttal document, the link between W tilde -> W hat is a quantization operation, which is non-differentiable. So, back-propagating gradients from the output of the main network to the meta quantizer suffers from the same problem of non-differentiability the authors so vehemently claim to have solved. Further, note that the rebuttal provided strongly disagrees with a key claim in the main paper on line 140: "Therefore, M_phi is connected to the final quantization training loss, which receives gradient update on phi backpropagated from the final loss... MetaQuant ***not only avoids the non-differentiability issue for the parameters**** in the model, but also...". Obviously, this backpropagation goes through a non differentiable step along the way (W tilde <- W hat) and so the problem is not solved, it is simply delegated from the main network to the meta-quantizer making the contribution void. I have therefore decided to decrease my score by one point (from 5 to 4).

Reviewer 3



Originality: the approach in the paper is novel and one that I haven't seen before. It provides an end-to-end training platform. I see it as essentially modeling a neural network to learn the residual dynamics that gets thrown away under the STE model. Quality: The paper is thorough. It motivates the problem well, does a good job of explaining its approach and lays out experiments on a number of networks and baselines. Clarity: The paper is clear in it's explanation of the problem, its approach and the results. Significance: Limited - since results are shown only on CIFAR benchmarks. It would be interesting to see results on Imagenet. Post-response comment: Thanks to the authors for pointing out their benchmark on ImageNet which I had overlooked during the original review. I have improved my score to 8.

Reviewer 4



In a weight-quantized network, the quantization function is usually non-differentiable. However, many methods need to use full-precision weight for update. Previous methods usually use heuristic methods to transform the gradient w.r.t. the quantized weights to full-precision weight. This paper proposes to learn a meta network to predict this transform. The authors also propose three ways to parameterize the meta network. The paper is overall easy to follow. My main concerns are in the experimental settings and results. 1. In line 203, the authors said that they report the "best test accuracy". This is not fair. 2. It is not clear that the proposed MetaQuant works under what kind of conditions. From the experiment results in Appendix A in the supplementary material. The proposed MetaQuant-FC sometimes has very poor performance. On the other hand, the previous STE methods, though might perform worse sometimes, can get relative stable performance for all the reported tasks. 3. In Figure 3 (b), when Adam is used, the STE method has similar or even better convergence behavior as the proposed MetaQuant at the early stage of training. However, it is run for a smaller number of iterations than the proposed MetaQuant. Thus it is hard to draw a comparison between these two kinds of methods. For fair comparison, all the competing methods should be run for the same number of iterations. It is also not clear whether the numbers reported in Tables 1-4 are run with the same number of iterations.

[Author Response · NeurIPS 2019]

**To Reviewer 1**: For question (1), the correlation among weights in kernels was indeed considered in the early design of
MetaQuant: the meta gradient of weights is determined by its surrounding weights and the gradient of its quantized
weight. However, such an implementation showed roughly 10-20% drop in performance compared to the current design.
Besides, the kernel sizes differ across layers in a deep neural network: a meta quantizer that receives $3 \times 3$ kernel as
inputs cannot be applied to the layers with $1 \times 1$ kernels. The independent process for each weight in the current design
endows MetaQuant with generalization to arbitrary architectures of neural networks.

For question (2), we further tested training time per iteration as suggested for MetaQuant and DoReFa with STE using
ResNet20 in CIFAR10 (Intel Xeon CPU E5-1650 with GeForce GTX 750 Ti). MetaQuant costs 51.15 seconds to finish
one iteration of training while baseline method uses 38.17s. We will add this training time analysis in final version.

For question (3), we further conducted experiments and added some state-of-the-art results of binary / ternary network
on ImageNet in Table 1:

| Network | Method | bits | Top 1/5 drop (%) | Network | Method | bits | Top 1/5 drop (%) | Network | Method | bits | Top 1/5 drop (%) |
|---|---|---|---|---|---|---|---|---|---|---|---|
| ResNet18 | MetaQuant | 1 | 6.32/4.31 | ResNet18 | MetaQuant | 2 | 5.17/3.59 | MobileNetV2 | MetaQuant | 4 | 2.10/0.38 |
| | STE* | 1 | 7.70/5.43 | | STE | 2 | 6.29/4.58 | | STE | 4 | 3.71/1.89 |
| | ELQ[1]** | 1 | 3.55/2.65 | | TTQ[2]*** | 2(Ternary) | 3.00/2.00 | | | | |

Table 1: Comparison experiments in ImageNet.*: The baseline methods in paper that use dorefa as forward and STE as backward. **: ELQ is a combination of a series of previous quantization methods and tricks on incremental quantization. MetaQuant focuses more on how to improve STE-based training quantization, without any extra loss and training tricks. ***: TTQ is a non-symmetric ternarization with $\{0, \alpha, -\beta\}$ as ternary points. MetaQuant follows dorefa using a symmetric quantization which leads to efficient inference.

**To Reviewer 2**: Regarding "in eq. (8) the term $\tilde{\mathbf{W}}$ ...", we
would like to clarify the computation order in a forward pass.
In fact, for an iteration $t$, $\tilde{\mathbf{W}}_t$ is computed after $\phi$ as shown
in the computation graph in Fig.2 of the paper. A more
detailed illustration is shown in Fig.1: The meta quantizer
$\mathcal{M}_\phi$ takes $\partial L/\partial \hat{\mathbf{W}}_{t-1}$ and $\tilde{\mathbf{W}}_{t-1}$ in the previous iteration
as inputs to compute its output, parameterized by $\phi$. The

Figure 1: Learning process of the meta quantizer.

output of meta quantizer is then added to $\mathbf{W}_{t-1}$ to generate $\tilde{\mathbf{W}}_t$ in the current iteration $t$. Therefore, $\partial \tilde{\mathbf{W}}/\partial \phi$ can be
computed if we track the path from $\phi$ to $\tilde{\mathbf{W}}_t$.

Regarding "... there seems to be a chicken-egg problem", the meta quantizer is actually linked to the final loss $L$ of the
base network with the following computation path: $\phi \to \Delta\mathbf{W} \to \tilde{\mathbf{W}} \to \hat{\mathbf{W}} \to L$, according to Fig.1. In detail: Step 1:
In iteration $t$, $\partial L/\partial \hat{\mathbf{W}}_{t-1}$ and $\tilde{\mathbf{W}}_{t-1}$ (noted they are from the previous iteration) are fed into **meta quantizer** as data
to generate meta gradient $\Delta\mathbf{W}$. Step 2: $\Delta\mathbf{W}$ contributes to $\tilde{\mathbf{W}}_t$, which is quantized to $\hat{\mathbf{W}}_t$. Step 3: $\hat{\mathbf{W}}_t$ is involved
into convolution or fully connected operation with input features from the base network, finally leads to the **loss**.
Intuitively, we can regard $\partial L/\partial \hat{\mathbf{W}}_{t-1}$ and $\tilde{\mathbf{W}}_{t-1}$ from previous iteration as data to meta quantizer ($\phi$) for generating a
component of $\hat{\mathbf{W}}_t$, and this process is differentiable. We can get $\partial L/\partial \hat{\mathbf{W}}_t$ using backpropagation, which can be passed
to $\phi$ by chain rules.

Regarding "... should the loss function of the base network be used for training ...", note that the goal of base network is
to minimize the final prediction loss while the aim of the meta quantizer is to provide accurate gradient $\partial L/\partial \hat{\mathbf{W}}$. Ideally,
the meta quantizer should be trained using 'ground-truth gradients' as regression values. However, such 'ground-truth
gradients' are inaccessible in practice. That's why STE is used to approximate the gradients in previous methods. In
order to train the meta quantizer without ground-truth values, we instead treat the final prediction loss of the base
network as indirect supervision. The final prediction loss could guide the meta quantizer towards reliable estimation for
'ground-truth gradients'. Therefore, in MetaQuant, the loss function of base network is used to train meta quantizer .
Empirically, this indirect training shows better performance and faster convergence than STE in our experiments.

For the writing issues, thanks for pointing then out. We will correct them in the final version.

**To Reviewer 3**: Thanks for your comments. In fact, we conducted experiments in ImageNet in **Table.3 on page 7** in
the original submission. Here, we have further conducted experiments, and added more state-of-the-art results of binary
/ ternary network on ImageNet in Table.1 for comparison.

# References

[1] Aojun Zhou, Anbang Yao, Kuan Wang, and Yurong Chen. Explicit loss-error-aware quantization for low-bit deep neural networks. In *Proceedings of the IEEE Conference on Computer Vision*
*and Pattern Recognition*, pages 9426–9435, 2018.

[2] Chenzhuo Zhu, Song Han, Huizi Mao, and William J Dally. Trained ternary quantization. *arXiv preprint arXiv:1612.01064*, 2016.


[Meta-Review · NeurIPS 2019]

This paper proposes to estimate the gradient w.r.t. the full-precision weight in a weight-quantized network by learning a meta network. The idea is original, and the paper is easy to follow overall. Hwoever, there is some concern on the experimental results and setup. Specifically, the authors provide three designs of meta quantizer, but they report the *best* test accuracy over these three. This may not be a fair comparison with the baseline STE. In the appendix, the authors reported detailed results for each design on cifar10 and cifar100, but not on imagenet. Also, apparently the authors use the change in loss values as stopping criterion. But as can be seen from figure3 (and other figures in the appendix), the loss values can still fluctuate a lot towards the end of training, and so this may again lead to unfair comparison.